# Radiation-Based Crosslinking Technique for Enhanced Thermal and Mechanical Properties of HDPE/EVA/PU Blends

**DOI:** 10.3390/polym13162832

**Published:** 2021-08-23

**Authors:** Jang-Gun Lee, Jin-Oh Jeong, Sung-In Jeong, Jong-Seok Park

**Affiliations:** Advanced Radiation Technology Institute, Korea Atomic Energy Research Institute, Jeongeup-si 56212, Korea; dlwkdrjs@kaeri.re.kr (J.-G.L.); nanjinoh@gmail.com (J.-O.J.); sijeong@kaeri.re.kr (S.-I.J.)

**Keywords:** polymer blending, radiation crosslinking, polyethylene, polyurethane, heat resistance, mechanical property

## Abstract

Crosslinking of polyolefin-based polymers can improve their thermal and mechanical properties, which can then be used in various applications. Radiation-induced crosslinking can be done easily and usefully by irradiation without a crosslinking agent. In addition, polymer blending can improve thermal and mechanical properties, and chemical resistance, compared to conventional single polymers. In this study, high-density polyethylene (HDPE)/ethylene vinyl acetate (EVA)/polyurethane (PU) blends were prepared by radiation crosslinking to improve the thermal and mechanical properties of HDPE. This is because HDPE, a polyolefin-based polymer, has the weaknesses of low thermal resistance and flexibility, even though it has good mechanical strength and machinability. In contrast, EVA has good flexibility and PU has excellent thermal properties and wear resistance. The morphology and mechanical properties (e.g., tensile and flexure strength) were characterized using scanning electron microscopy (SEM) and a universal testing machine (UTM). The gel fraction, thermal shrinkage, and abrasion resistance of samples were confirmed. In particular, after storing at 180 °C for 1 h, the crosslinked HDPE-PU-EVA blends exhibited ~4-times better thermal stability compared to non-crosslinked HDPE. When subjected to a radiation dose of 100 kGy, the strength of HDPE increased, but the elongation sharply decreased (80%). On the other hand, the strength of the HDPE-PU-EVA blends was very similar to that of HDPE, and the elongation was more than 3-times better (320%). Finally, the abrasion resistance of crosslinked HDPE-PU-EVA was ~9-times better than the crosslinked HDPE. Therefore, this technology can be applied to various polymer products requiring high heat resistance and flexibility, such as electric cables and industrial pipes.

## 1. Introduction

Polyethylene, in particular, is the most representative general-purpose polymer among the polyolefin-based polymers. It exhibits properties such as crystallinity and a decrease in crystal size depending on the number and length of branches connected to the main chain, and variations can be divided into HDPE and low-density polyethylene (LDPE). Among them, HDPE has a small number of short and/or long molecular chains, has a high crystallinity (60 to 80%), and has a melting point of 135 °C [1,2,3,4]. In addition, HDPE is used in diverse ways, such as film, blow molding, injection molding, pipe processing, and wires due to its many advantages (high tensile strength, excellent processability, excellent low-temperature resistance, and electrical insulation) [5,6,7]. However, HDPE has problems with rapid mechanical degradation and thermal shrinkage at above 130 °C near the melting point [8,9]. To overcome these disadvantages, polymer blends are prepared by blending and/or crosslinking to improve their mechanical properties [10,11,12,13]. In our previous study, improved thermal and mechanical properties were confirmed after blending styrene-grafted polyurethane (PU) into polypropylene (PP), a polyolefin-based polymer. PU is a non-toxic polymer with excellent elasticity, abrasion resistance, and workability. It was confirmed that the original mechanical properties (disadvantageous low-temperature characteristics) of PP were improved. In this way, mechanical properties can be improved through polymer blending [14]. 

In addition, it is possible to improve further or maintain the thermal and mechanical properties of polymer blends through polymer crosslinking [15]. Crosslinking refers to the connection by physical and chemical interactions between polymer chains. Polymer crosslinking can be divided into physical and chemical crosslinking [16]. Physical crosslinking is a crosslinking method involving interactions such as the entanglement of chains or ionic bonds between chains. Chemical crosslinking is provided by chemical bonds (such as covalent bonding) between molecular chains [17,18,19,20,21,22,23]. In particular, among the chemical crosslinking methods, the radiation crosslinking method has the advantage that a reaction is induced without the use of a chemical additive, such as a crosslinking agent or initiator. Moreover, a reaction can be induced at various temperatures and in various states (e.g., solid, liquid, and gas) [24,25,26,27]. The crosslinking of HDPE using radiation crosslinking technology is useful for generating radicals in the HDPE polymer chain. HDPE has the characteristic that when tertiary carbon atoms in the branch chain lose hydrogen, they generate radicals that are easily attacked by external free radicals and stabilized. Thus, crosslinking occurs readily due to easy radical generation [28,29]. 

In this study, an HDPE blend was prepared by blending ethylene vinyl acetate (EVA) and PU to improve their thermal and mechanical properties. EVA is a copolymer of ethylene and vinyl acetate (VA), and its density, flexibility, elasticity, and durability vary depending on the VA content. In addition, EVA is widely used as an over wire cable and as a solar sheet for its excellent low-temperature characteristics and impact resistance [30,31]. EVA and PU, which have these advantages, were blended with HDPE to prepare a blend and then crosslinked using electron beam irradiation. To determine the properties of the crosslinked HDPE/EVA/PU (H/V/U) blend, the gel fraction, shrinkage rate, tensile strength, flexural strength, and wear resistance were confirmed. We intend to confirm the improvement of the thermal and mechanical properties of this blend after electron beam crosslinking.

## 2. Materials and Methods

### 2.1. Materials

High-density polyethylene (HDPE) and ethylene-vinyl acetate (EVA) were purchased from the Lotte Chemical Corporation (Seoul, Korea). Polyurethane (PU) was obtained from Songwon Industrial Co., Ltd. (Ulsan, Korea). Polyethylene-graft-maleic anhydride (PE-g-MA) (viscosity of 500 cP) was purchased from Sigma-Aldrich (St. Louis, MI, USA). All other reagents and solvents were of analytical grade and used as received. 

### 2.2. Preparation of the HDPE-EVA-PU Blends 

HDPE, EVA, and PU were blended using a Brabender mixer (Brabender D-47055, Brabender, Duisburg, Germany) at 45 rpm and 190 °C for 20 min. HDPE-EVA-PU blend sheets were prepared using a hot press at 190 °C. To achieve the crosslinking, the sheets were exposed to electron beam irradiation (2.5. MeV, UELV-10-105, Korea Atomic Energy Research Institute, Jeongeup, Korea) to achieve doses of 50, 75, and 100 kGy (25 kGy/cycle).

### 2.3. Characterization of the HDPE-EVA-PU Blends

The surface morphology of the samples was observed using scanning electron microscopy (SEM, TM3030, HITACHI, Tokyo, Japan). To acquire high-resolution images, the samples were coated with gold for 60 s using sputter coating to conduct SEM with a 15 kV electron beam and a working distance of 8.1 mm. In addition, chemical component and quantitative analyses were performed using energy-dispersive X-ray spectroscopy (EDS, TM3030, HITACHI, Tokyo, Japan).

To analyze the gel fraction of the HDPE-EVA-PU blends using Soxhlet extraction, samples (0.5 × 0.5 cm^2^) were prepared, and their initial weights were recorded prior to immersion in xylene for 8 h at 140 °C to remove unreacted polymers. After that, the samples were dried in an oven at 80 °C for 8 h and then dried naturally for 4 h after which the weight was recorded. The gel fraction was calculated from the following equation:Gel fraction (%) = (*W_f_*/*W_i_*) × 100
where *W_i_* and *W_f_* represent the initial and final weights of the dried samples. 

Prior to testing for tensile strength, the samples were prepared according to ASTMD638 and measured using a universal testing machine (UTM, Instron 5982, Norfolk, MA, USA) with a 100 kN range of load and crosshead speed of 50 mm/min. In addition, the flexural strength of samples (8 × 1 cm^2^) was measured using the UTM with a 10 kN range of load, 48 mm of span distance, and 50 mm/min of crosshead speed. 

To confirm the thermal shrinkage, samples were prepared (4 × 4 cm^2^) and placed in an oven at 150 or 180 °C for 1 h before the result was recorded. The thermal shrinkage of samples was calculated using the following equation:Thermal shrinkage (%) = [1 − (*S_a_*/*S_b_*)] × 100
where *S_a_* and *S_b_* represent the thermal shrinkage after and before tested area, respectively.

The abrasion resistance of the samples was measured according to KSD8314 (1000 cycles, speed 60 rpm, test load of 1 kg, stroke of 50 ± 1 mm at room temperature). The abrasive material was sandpaper (Koptri, #220). The weight reduction rate of the samples was calculated using the following equation:Weight reduction rate (%) = [(*T_a_*− *T_b_*)/*T_a_*] × 100

*T_a_* is the weight before the test and *T_b_* is the weight after the test.

## 3. Results and Discussion

### 3.1. Preparation of HDPE/EVA/PU Blends

Polymer blending is one of the methods that can be used to supplement the weaknesses of a polymer, and properties can be enhanced, and desired properties can be manufactured through blending [32,33,34]. In this study, HDPE, EVA, and PU were blended using a Brabender mixer to prepare H/V/U blend sheets using a hot press. To crosslink each sample, the H/V/U blends were irradiated using electron beam irradiation at different doses (50, 75, and 100 kGy). The overall schematic illustration and chemical composition of the H/V/U blends are shown in Figure 1 and Table 1.

### 3.2. Characterization of HDPE/EVA/PU Blends

HDPE is composed of pure carbon and hydrogen and exhibits non-polar properties [14]. EVA is a copolymer of ethylene and VA and is also non-polar, whereas VA contains oxygen atoms and exhibits polar properties [30,31]. Therefore, EVA mixes well with non-polar polymers and polar polymers and has the property that it can easily be mixed with various kinds of chemical additives. In addition, because PU also exhibits non-polar properties, HDPE, EVA, and PU are considered to blend effectively. Figure 2 shows the results of the surface morphology and elemental composition of H, H/U, H/V, and H/V/U determined using SEM/EDS. As shown in Figure 2a, as a result of the surface resulting after preparing a sheet by blending each polymer using a Brabender mixer and a hot press, a flat surface was confirmed in all samples. In addition, it was confirmed that the form of separation between polymers was not significant. In addition, Figure 2b shows the EDS results for each element (green: carbon, blue: nitrogen, and red: oxygen). The Nitrogen (blue) of PU was confirmed in H/U and H/V/U, and the oxygen (red) of EVA and PU was confirmed in H/U, H/V, and H/V/U. It was also confirmed that each element was uniformly distributed in the blends, which showed effective blending. 

Figure 3 shows the gel fraction of H and the H/V and H/V/U blends after electron beam irradiation. As shown in Figure 3a, the gel fraction of each sample was confirmed according to the radiation dose of 100 kGy. In the case of H, a 58% gel fraction was confirmed, and the gel fraction of the H/V and H/V/U blends was higher (66 and 65%, respectively). This result was believed to be due to crosslinking of HDPE as well as EVA. In general, the mechanical properties of EVA could be improved by crosslinking using peroxide (e.g., dicummyl peroxide or perbutyl peroxide) because EVA has weaker mechanical properties than PE as applied to polyethylene and polyester [35]. In this study, it was confirmed that the gel fraction was increased through crosslinking of HDPE and EVA using only electron beam irradiation (that is, without the use of a crosslinking agent). In the case of H/U blends containing PU, the gel fraction was not significant compared to that of H/V. This result is believed to be because EVA was crosslinked by electron beam irradiation more efficiently than PU. In addition, the gel fraction of each sample was confirmed by the dose of electron beam irradiation. Figure 3b shows that the gel fraction result was confirmed by the different radiation doses of HDPE. The gel fraction increase in relation to the radiation dose of 50, 75, and 100 kGy is 50, 56, and 58%, respectively. The gel fraction of the H/V blend increases to 55, 61, and 66%, respectively, as the radiation dose increases to 50, 75, and 100 kGy (shown in Figure 3c). In addition, the increase in the gel fraction of the H/V/U blend with the dose of 50, 75, and 100 kGy is 54, 65, and 65%, respectively.

As shown in Figure 4, the mechanical properties of the samples were confirmed through crosslinking by electron beam irradiation, and Figure 4a shows the tensile stress of each sample. In the case of the tensile stress of H and the H/U blends, it was confirmed that the tensile stress was only slightly changed, even if the radiation dose was increased from 50 to 100 kGy. In particular, it was confirmed that the tensile stress of the H/V and H/V/U blends decreased when the radiation dose was 100 kGy. Although the tensile stress of the H/V blends increased with an increase in the radiation dose from 0 kGy (25 MPa) to 50 kGy (25.3 MPa) and 75 kGy (27 MPa), at the dose of 100 kGy, the tensile stress decreased to 23 MPa. In addition, the tensile stress of the H/V/U blends increased with an increase in the radiation dose but decreased to 23 MPa in the case of 10 kGy (like for the H/V blends). Figure 4b shows the result of the tensile strain of each sample. With the increase in radiation dose from 0 to 50, 75, and 100 kGy, the HDPE strain decreases to 510%, 370%, 100%, and 80%. In particular, it was confirmed that the strain decreased rapidly at 75 and 100 kGy. This result is believed to be because the tensile strain decreases as the amount of crosslinking in HDPE was increased with an increase in the radiation dose. In the case of the tensile strain of the H/U blend, the tensile strain decreased slightly (compared to HDPE) because PU has excellent mechanical properties. The H/V and H/V/U blends with EVA exhibited higher tensile strain than with HDPE at 100 kGy. Moreover, it was confirmed that the mechanical properties were improved. The H/V blends decreased to 570%, 500%, 450%, and 390% as the radiation dose increased from 0 to 100 kGy. In the case of the H/V/U blends, the strain in specimens with radiation doses of 0, 50, 75, and 100 kGy was decreased to 530%, 480%, 410%, and 320%. The tensile strains of H and the H/V/U blends with a radiation dose of 100 kGy were 80% and 320%, confirming that the strain of the H/V/U blends with EVA and PU increased approximately four times. This result shows that the H/V/U blend exhibits excellent tensile strength.

Flexure strength is the resistance to the bending of various materials such as plastic, ceramic, and rubber samples. It is defined as the maximum value at which the load does not increase anymore when applying a bending force. Figure 5a shows the flexure stress of each sample. The flexure stress of HDPE is 27, 30, 29, and 29 MPa at the radiation dose of 0, 50, 75, and 100 kGy, and the flexure stress of the H/U blend is 33, 29.7, 30.2, and 29 MPa as the radiation dose increases to 0, 50, 75, and 100 kGy. Even after crosslinking by electron beam irradiation, there is only a slight change in the flexure stress. On the other hand, the flexure stress of the H/V and H/V/U blends is lower than that of the H and the H/U blend. It was confirmed that there was only a slight change, even when the radiation dose increased. The flexure stress of the H/V blend was 22.5, 22.3, 23, and 22.9 MPa according to the radiation dose of 0, 50, 75, and 100 kGy. In addition, the flexure stress of the H/V/U blend was 21.5, 22, 21.3, and 21.4 MPa at the radiation dose of 0, 50, 75, and 100 kGy. This result shows that the flexure stress was lowered by blending with EVA. Figure 5b shows the flexure strain of each sample. The flexure strain of HDPE decreased after crosslinking by electron beam irradiation. The flexure strain of HDPE was 19.8, 14.7, 14, and 13.8% according to the radiation dose of 0, 50, 75, and 100 kGy. On the other hand, it was confirmed that the flexure strain of the H/U blend blended with PU increased to 19.5, 20.7, 22.5, and 24% at the radiation dose of 0, 50, 75, and 100 kGy. This result is believed to be because the flexure strain increased due to the excellent tensile properties of PU. In addition, the flexure strains of the H/V and H/V/U blends in which EVA was blended were also confirmed to have similar tendencies by the inclusion of PU (as in the result for H and the H/U blend). The flexure strain of the H/V blend was 19.9, 17.3, 16.1, and 16.8% according to the radiation dose of 0, 50, 75, and 100 kGy, and the flexure stain of the H/V/U blend was determined by the radiation dose of 0, 50, 75, and 100 kGy (with the results of 21%, 24%, 24.8%, and 25.1%).

The physical properties (gel fraction, tensile, and flexure strength) were also confirmed by the ANOVA one-way (Table 2).

Polyethylene (PE) begins to collapse when the crystal structure is above 70 ℃, which increases the amorphous state. The melting point is where it becomes 100% amorphous. The melting point of HDPE is 132–138 ℃, the heat of fusion is 55–66 cal/g, and the melting point changes depending on the copolymerized branches and molecular weight. In addition, the glass transition phenomenon in HDPE, a semi-crystalline polymer, occurs by the thermal behavior of the amorphous region between the crystalline lamellae. The glass transition phenomenon determines the fragility of the polymer according to the thermal deformation temperature. The glass transition temperature (Tg) is lowered due to the cooling rate of HDPE and having more chain branches and is increased by the higher molecular weight and intermolecular crosslinking [36,37,38]. In particular, the thermal conductivity of HDPE is caused by the complex vibration of atoms in the lattice structure. For this reason, the thermal conductivity of HDPE with high crystallinity is very high compared to other polyolefins with low crystallinity, such as LDPE or PP. The thermal conductivity of HDPE is strongly proportional to its density, decreases with increasing temperature, and increases with increasing pressure [39,40,41]. In particular, it is very important to consider the thermal conductivity of HDPE when preparing thick HDPE moldings. Figure 6 shows the thermal shrinkage results at 150 °C and 180 °C for H, and the H/U, H/V, and H/V/U blends with radiation doses at 0 and 75 kGy. As shown in Figure 6a, in the case of non-irradiated samples, the confirmed reduction rate of HDPE is 9.8%, and the H/U, H/V, and H/V/U blends are 6.7%, 9.3%, and 6.2%, respectively. On the other hand, in the case of the sample irradiated with 75 kGy, it was confirmed that the reduction rate was reduced by 3.8%, 0.1%, 2.9%, and 1.8%, respectively, compared to non-irradiated samples. In particular, in the case of the H/U blend, it was confirmed that thermal shrinkage hardly occurred after electron beam irradiation. This result suggests that thermal shrinkage did not occur due to the excellent thermal properties of PU. These results were the same as the thermal shrinkage results of PP blended and styrene-grafted PU blends in previous studies. To improve the thermal properties of PP, after blending PU with grafted styrene, it was left at 150 °C for 1 h to confirm from the shape of samples that there was almost no change. In addition, in the case of the H/V/U blends, it was possible to confirm the result of crosslinking of HDPE by electron beam irradiation. 

After efficient blending of EVA and PU, a reduction rate of 1.8% was confirmed after irradiation at 75 kGy. In Figure 6b, the result of thermal shrinkage showed in samples after storage at 180 °C for 1 h. In cases of non-irradiated samples, the reduction rate of HDPE was confirmed to be 11.9%, and reduction rates of 9.1%, 16%, and 9.7% were confirmed for the H/U, H/V, and H/V/U blends, respectively. In addition, in the case of the sample irradiated at 75 kGy, it was confirmed that the shrinkage rate for the H, H/U, H/V, and H/V/U blends was reduced by 4.2%, 2.1%, 4.1%, and 2.8%, respectively. With non-irradiated H/V/U blends, in particular, it was confirmed that the thermal shrinkage phenomenon rapidly increased as the temperature increased from 150 to 180 °C. It is considered that the poor thermal properties of HDPE at high temperatures affect the thermal shrinkage results. Based on these results, the possibility of improving the thermal properties of plastic polymers was confirmed through the easy and useful blending of polymers and crosslinking using electron beam irradiation. Figure 6c shows an optical image of the thermal shrinkage of each sheet-type sample.

Abrasion resistance refers to the property of resisting abrasion well, and it is applied by using abrasion-resistant materials for accessory parts for such as electric wires, automobiles, and airplanes. For example, regarding wires, if its cover is not abrasion-resistant, it could cause a major fire due to the peeling of the insulation off the wire over time. Thus, if the stability of the material decreases and causes a serious risk of personal injury or accident, this is a big disadvantage [42,43]. Therefore, abrasion resistance is an important factor for plastic materials. Figure 7 shows the results of tests pf abrasion resistance of HDPE and the crosslinked samples irradiated at 100 kGy using electron beam irradiation. With non-irradiated HDPE (H 0), a weight reduction rate of 0.08% was confirmed, but in the case of HDPE irradiated at 100 kGy, a high weight reduction rate of 0.27% was shown. In particular, it was confirmed that crosslinked HDPE had weak abrasion resistance. On the other hand, when the radiation dose was 100 kGy, in the H/U (H/U 100), H/V (H/V 100), and H/V/U (H/V/U 100) blends with EVA and PU, excellent abrasion resistance properties were indicated by the low weight reduction rates of 0.02, 0.04, and 0.03%.

## 4. Conclusions

In this study, H/V/U blends were prepared by radiation crosslinking to improve the thermal and mechanical properties of HDPE. In the case of H/V/U blends with radiation doses of 75 and 100 kGy, the mechanical properties, and thermal shrinkage were sharply enhanced compared to non-irradiated HDPE. After storage at 180 °C for 1 h, the crosslinked HDPE/PU/EVA blends exhibited about 4-times better thermal stability compared to non-crosslinked HDPE. In addition, when the radiation dose was 100 kGy, the strength of HDPE increased, but the elongation sharply decreased (80%). On the other hand, the strength of the HDPE-PU-EVA blends was similar to that of HDPE, and the elongation was >3-times better (320%). Finally, the abrasion resistance of the crosslinked HDPE/PU/EVA was ~9-times better than with crosslinked HDPE. These excellent properties of the H/V/U blends can be used not only for insulated cable, thermal shrinkage tube, and pipes but also in automobile components. 

## Figures and Tables

**Figure 1 polymers-13-02832-f001:**
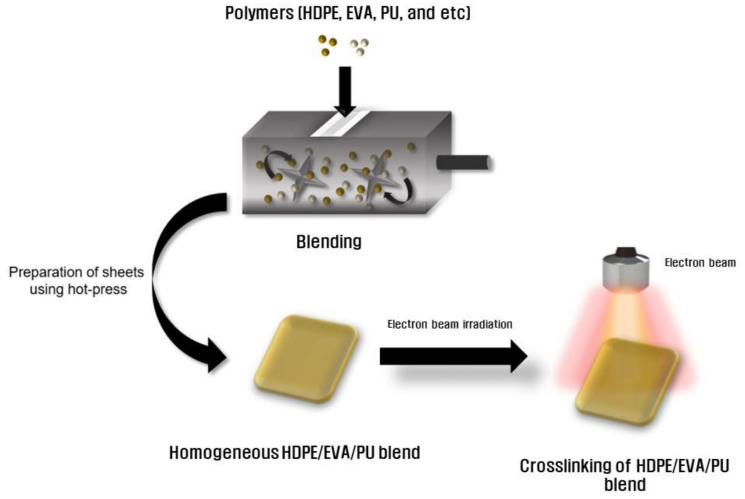
Schematic illustration of the HDPE/EVA/PU blends after using electron beam irradiation.

**Figure 2 polymers-13-02832-f002:**
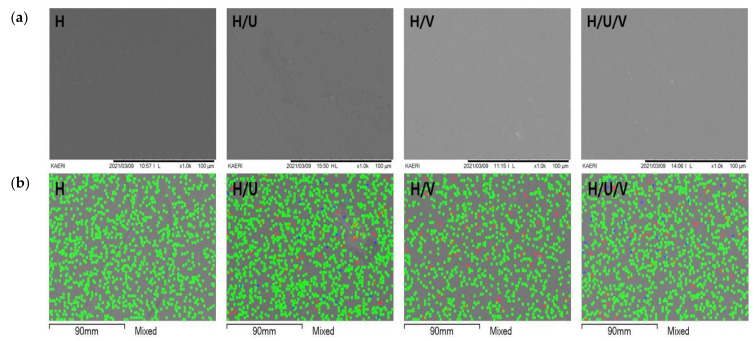
Surface morphology and elemental composition of the HDPE/EVA/PU blends by (**a**) scanning electron microscopy and (**b**) energy-dispersive X-ray spectroscopy.

**Figure 3 polymers-13-02832-f003:**
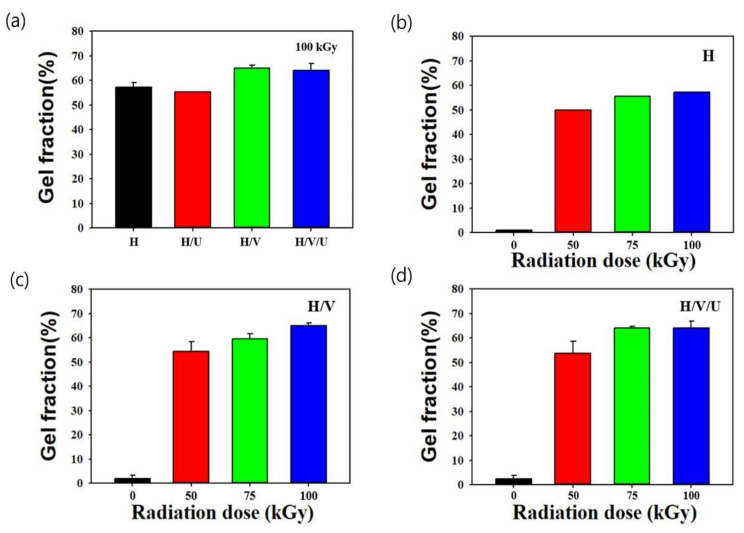
Gel fraction of the HDPE/EVA/PU blends: (**a**) radiation dose at 100 kGy, (**b**) different radiation doses at 0, 50, 75, and 100 kGy in HDPE, (**c**) different radiation doses at 0, 50, 75, and 100 kGy in HDPE/EVA blends, and (**d**) different radiation doses at 0, 50, 75, and 100 kGy in HDPE/EVA/PU blends.

**Figure 4 polymers-13-02832-f004:**
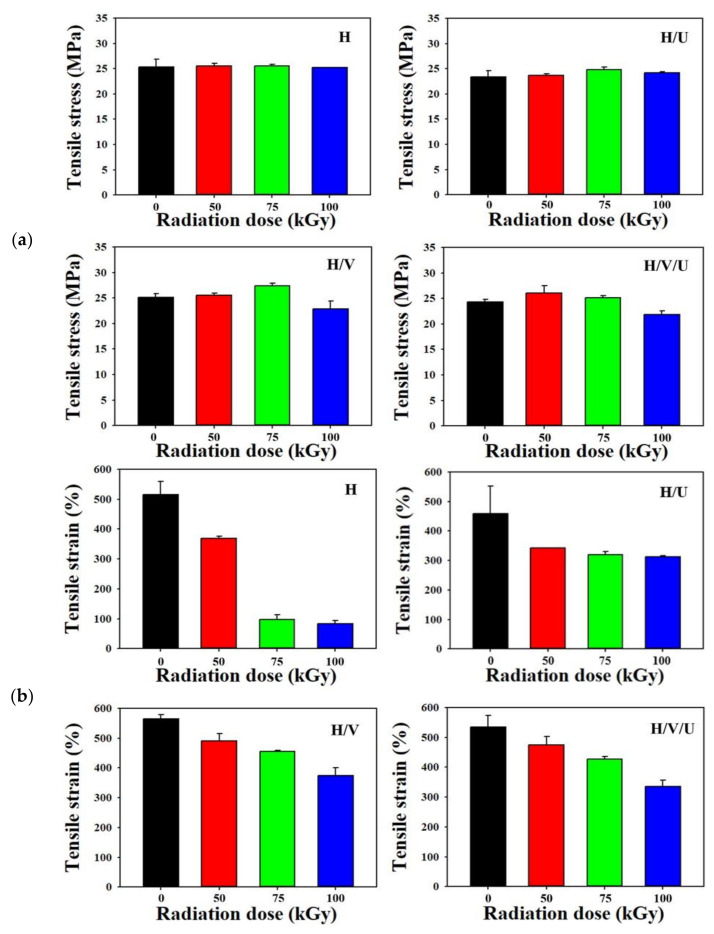
Tensile properties of HDPE/EVA/PU blends: (**a**) tensile stress of each sample with different radiation dose, (**b**) tensile strain of each sample with different radiation dose.

**Figure 5 polymers-13-02832-f005:**
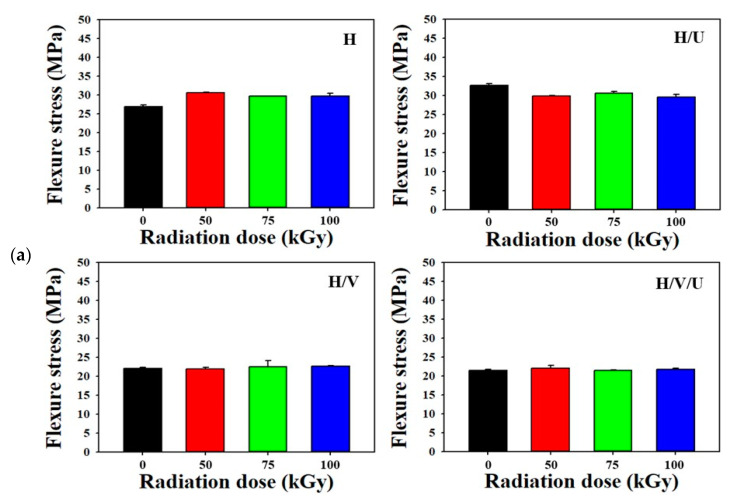
Flexure properties of the HDPE/EVA/PU blends: (**a**) flexure stress of each sample with different radiation dose, (**b**) flexure strain of each sample with different radiation dose.

**Figure 6 polymers-13-02832-f006:**
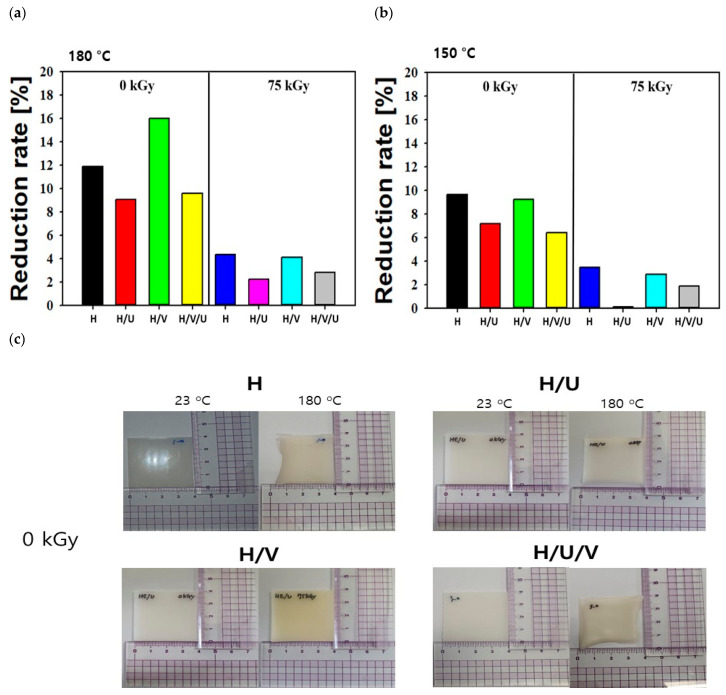
Thermal shrinkage of HDPE/EVA/PU blends: (**a**) reduction rate of non-irradiated sample and samples irradiated at 75 kGy and 150 °C for 1 h, (**b**) reduction rate of non-irradiated sample and samples irradiated at 75 kGy and 180 °C for 1 h, and (**c**) optical images of the HDPE/EVA/PU blends with non-irradiated and samples irradiated at 23 °C and 180 °C for 1 h.

**Figure 7 polymers-13-02832-f007:**
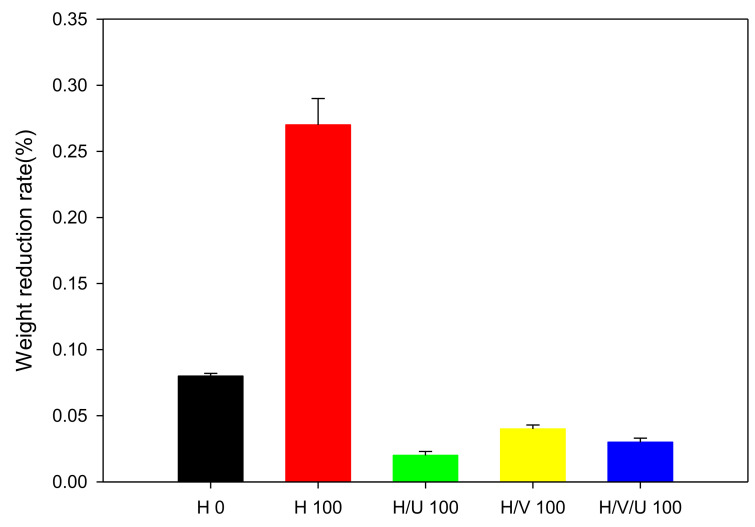
Abrasion resistance of HDPE/EVA/PU blends.

**Table 1 polymers-13-02832-t001:** Chemical composition of the HDPE/EVA/PU blends.

	H	H/U	H/V	H/V/U
HDPE	200 g	200 g	200 g	200 g
EVA	-	-	20 phr	20 phr
PU PE-g-MA	- -	5 phr 3 phr	- 3 phr	5 phr 3 phr

**Table 2 polymers-13-02832-t002:** ANOVA table of physical properties.

Physical Property	F-Value	*p*-Value	F Crit
Gel fraction	0.04	0.96	4.26
Tensile stress	1.97	0.17	3.49
Tensile strain	2.21	0.14	3.49
Flexure stress	112.43	0.00	3.49
Flexure strain	9.46	0.00	3.49

## Data Availability

The data presented in this study are available on request from the corresponding author.

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
