# Peer review of "Radiation-Based Crosslinking Technique for Enhanced Thermal and Mechanical Properties of HDPE/EVA/PU Blends"

_polymers, 2021, doi:10.3390/polym13162832_

Round 1

Reviewer 1 Report

Dear authors,
Thank you very much for this practical and useful scientific article.

The article is very well prepared, the whole concept, information and results are at a high level.

Although the article is prepared well, I recommend a few technical notes:
- the images are insufficient in these sizes - this is especially true for the resulting graphs and photographs,
- The first time you use the acronym H / V / U, it should be clear what it is (although it may sound negligible) - it could be confusing.
- I recommend expanding the description of the use of these materials in practice,

Regards,

Author Response

All comments and suggestions are greatly appreciated by authors since these suggestions and comments help us improve this manuscript. We have revised the manuscript carefully according to Reviewer's comment.

Point 1. The images are insufficient in these sizes - this is especially true for the resulting graphs and photographs.

Response 1: Thanks very much for your valuable opinion. We reorganized by adjusting the quality and size of figures in this paper to enhance the reader’s understanding.

Point 2. The first time you use the acronym H / V / U, it should be clear what it is (although it may sound negligible) - it could be confusing.

Response 2: Thanks very much for your valuable opinion. When we first use the abbreviation H/V/U in the text, we indicated what it is in the introduction part.

  • To determine the properties of the crosslinked HDPE/EVA/PU (H/V/U) blend, the gel fraction, shrinkage rate, tensile strength, flexural strength, and wear resistance were confirmed.

Point 3. I recommend expanding the description of the use of these materials in practice.

Response 3: Thanks very much for your valuable opinion. In the conclusion part, we mentioned about the products that can be applied as the features of this technology.

  • These excellent properties of the H/V/U blends can be used not only for insulated cable, thermal shrinkage tube, pipes but also in automobile components

Reviewer 2 Report

Overall is an excellent presented and discussed paper. The scope of the study is clearly given, the methodology is sound and the results are well discussed. However, I would like to see a statistical analysis on (ANOVA one way) the data presented in figures 3 to 7. This will help the authors to identify potential significant differences among the treatments. Therefore the paper can be accepted after minor revision. 

Author Response

All comments and suggestions are greatly appreciated by authors since these suggestions and comments help us improve this manuscript. We have revised the manuscript carefully according to Reviewer's comment.

Point 1. I would like to see a statistical analysis on (ANOVA one way) the data presented in figures 3 to 7. This will help the authors to identify potential significant differences among the treatments.

Response 1: Thanks very much for your valuable opinion. We performed ANOVA one way analysis on the gel fraction, tensile stress, tensile strain, flexure stress, and flexure strain as shown in Table 2.
